# Estimates of Dietary Exposure to Antibiotics among a Community Population in East China

**DOI:** 10.3390/antibiotics11030407

**Published:** 2022-03-17

**Authors:** Yingying Wang, Xinping Zhao, Jinxin Zang, Yurong Li, Xiaolian Dong, Feng Jiang, Na Wang, Lufang Jiang, Qingwu Jiang, Chaowei Fu

**Affiliations:** 1NHC Key Laboratory of Health Technology Assessment, School of Public Health, Fudan University, Shanghai 200032, China; 17211020095@fudan.edu.cn (Y.W.); xpzhao@shmu.edu.cn (X.Z.); 19261020025@fudan.edu.cn (J.Z.); 15621175797@163.com (Y.L.); jf98@fudan.edu.cn (F.J.); jianglufang@fudan.edu.cn (L.J.); jiangqw@fudan.edu.cn (Q.J.); 2Deqing County Center for Disease Prevention and Control, Huzhou 550004, China; djq04@126.com

**Keywords:** antibiotics, dietary exposure, foods, drinking water, East China

## Abstract

Background: Antibiotics are widely used in clinics, livestock farms and the aquaculture industry. A variety of antibiotics in foods and drinking water may lead to important and inadvertent dietary exposure However, the profile of dietary exposure to antibiotics in humans is not well-explored. East China is an economically developed area with a high usage of antibiotics and a high rate of antibiotic resistance (ABR). This study aimed to evaluate the total intake level of antibiotics in humans via foods and drinking water based on a community population in East China. Methods: A total of 600 local residents from 194 households were recruited into this study in Deqing County of Zhejiang Province since June 2019. Each subject was asked to fill a food frequency questionnaire to report their daily consumption of foods and drinking water. Tap water samples were collected from ten households and twenty-one antibiotics of five categories were selected to detect in drinking water. Data of antibiotic residues in animal-derived foods were obtained from the notification of unqualified edible agricultural products after special supervision sampling inspection in Deqing County. The human dietary exposure to antibiotics was estimated by combining the data of antibiotic contamination in foods and drinking water, and the information of dietary consumption. Results: Of twenty-one antibiotics selected, subjects were exposed to a total of sixteen antibiotics, ranging from 15.12 to 1128 μg/day via two main dietary routes (animal-derived foods and drinking water). The overall dietary exposure level varied greatly in the antibiotics detected and their sources. Compared with other antibiotics, enrofloxacin made the most contributions in terms of dietary exposure, with a median exposure level of 120.19 μg/day (IQR: 8.39–698.78 μg/day), followed by sulfamethazine (median: 32.95 μg/day, IQR: 2.77–162.55 μg/day) and oxytetracycline (median: 28.50 μg/day, IQR: 2.22–146.58 μg/day). The estimated exposure level via drinking water (at the ng/day level, median: 26.74 ng/day, IQR: 16.05–37.44 ng/day) was significantly and substantially lower than those via animal-derived foods (at the μg/day level, median: 216.38 μg/day, IQR: 87.52–323.00 μg/day). The overall dietary exposure level also showed differences in sex and age. Males and youths were more likely to be exposed to antibiotics via dietary routes than others. Conclusions: The community population investigated in East China was extensively exposed to multiple antibiotics via dietary routes. Long-term exposure to low-dose antibiotics in animal-derived foods was the primary dietary exposure route, compared with drinking water. Enrofloxacin contributed to the major body burden of dietary exposure, based on the combination of consumption of aquatic products and considerable enrofloxacin residues in them. Although the human dietary exposure level to antibiotics via drinking water and animal-derived foods ranged from ng/day to μg/L, their chronic toxicity and the accumulation and spread of ABR may be potential hazards to humans. Therefore, long-term monitoring of antibiotic contaminations in foods and drinking water, and human dietary antibiotic exposure is warranted.

## 1. Introduction

Recently, due to overuse and misuse in clinics, livestock farms and the aquaculture industry, the benefits of antibiotics on disease treatment and/or growth promotion have been challenged by their hazards for human health and ecological environment, which has also become a public concern [1]. The risk of antibiotics in human health generally manifests in the following two ways: adverse drug reaction (ADR) and the potential spread of antibiotic resistance (ABR) due to selective pressure on bacteria. One of such ADRs is called chronic toxicity, describing the circumstance under which antibiotics accumulate in the human body and then cause organ lesions due to low dose consumption [2].

China produces and uses significant quantities of antibiotics annually [3]. In 2013, the total amount of antibiotics used was as high as 162,000 tons, of which 48% was consumed by humans and the rest by livestock [4]. In 2018, the percentage of antibiotics used in inpatients was up to 40.4% [5]. These antibiotics end up in various waste water and livestock wastes. However, the limited infrastructure leads to a low treatment rate for sewage in China, especially in the majority of rural areas [6]. Additionally, unlike domestic wastewater, China has not yet implemented specific requirements for treating livestock wastes before discharge. East China is one of the most economically developed regions in China, housing more than 400 million inhabitants. With the rapid development of the pharmaceutical industry and intensive breeding industry, and a large number of large-scale hospitals, East China has become an important region affected by the significantly increasing consumption of antibiotics [4], and has become the hot spot for controlling antibiotic use and ABR [7,8].

With the inappropriate use of antibiotics in animal husbandry and the aquaculture industry, antibiotic residues have been frequently detected in drinking water samples and food samples (including meat, milk and aquatic products) [9,10]. Previous studies among children and adults in Shanghai have suggested that most of the antibiotics detected in urines were veterinary antibiotics or were preferred for use in animals, indicating that the exposure routes to humans by antibiotics is not limited to clinical use, but also derives from foods and drinking water [11,12]. Antibiotic residues cannot be completely eliminated by traditional Chinese cooking methods [13]. The gradual accumulation of antibiotics in the environment has further resulted in the contamination of foods and drinking water, ultimately transferred to humans during the food chain and accumulate in the human body. 

Considering the potential hazards to human health caused by antibiotic exposure, the Food and Agriculture Organization of the United Nations (FAO) and the World Health Organization (WHO) have developed and constantly updated limit standards of antibiotic residues in foods [14]. The health risks from dietary antibiotic exposure are not only related to the level of antibiotics in foods and drinking water, but also closely connected with the dietary patterns and daily dietary consumption of Chinese residents. Over the past decades, China has seen an ongoing shift of dietary patterns, which is mainly characterized by an increases in the consumption of animal-derived foods, with pork being most popular [15]. 

In previous studies, the estimation of human exposure to antibiotics was generally evaluated only by summary prescription information, which ignored other possible exposure routes, including the consumption of contaminated foods and drinking water, so the actual body burden due to antibiotic exposure might have been underestimated in humans [16,17]. However, comprehensive assessments on dietary exposure to antibiotics in humans are limited. In this study, based on a community population located in East China, we aimed to evaluate the total intake level of antibiotics in humans via dietary sources, and further identify the main antibiotics in the exposure spectrum.

## 2. Material and Methods

### 2.1. Study Design and Subjects

Based on the population cohort which had been previously established in Deqing County of Zhejiang Province, which is located in Taihu basin of Yangtze Delta, China, one community was randomly sampled in this study. A total of 600 local residents from 194 households (aged 18 years or above, living in the study regions for more than 6 months and having no plan to move out) were recruited in this study in June 2019. Those with serious body diseases such as cardiovascular and cerebrovascular diseases and liver and kidney diseases were excluded. Each subject was required to fill out a questionnaire regarding their demographic and social-economic characteristics, and was asked to complete a food frequency questionnaire regarding the consumption of drinking water and common foods. Drinking water samples were collected at the same time. Finally, 523 subjects with the completed data of dietary consumption were eligible for the final analysis. The response rate was 87.2%. A flow chart of the process is presented in Figure 1. Written informed consent was obtained from all subjects before data collection. This study was reviewed and approved by the Institutional Review Board of Fudan University School of Public Health (number: 2019-03-0733, date: 18 March 2019). 

### 2.2. Collections for Information of Consumption of Drinking Water and Animal-Derived Foods

According to the antibiotic residues in animal-derived foods reported in the previous study, in Refs. [10,18], the food frequency questionnaire covered a total of forty-two items from eleven food categories, including livestock meats (pork, beef, lamb), poultry meats (chicken, duck, goose), animal offal (gizzard, liver, kidney, heart, lung, blood), processed meats (bacon, sausage, luncheon meat), freshwater fishes (carp, crucian, chub, weever, snakehead, catfish, butterfish), marine fishes (hairtail, large yellow croaker, small yellow croaker), shrimps (river shrimp, small-sized shrimp, prawn, cray), crabs (swimming crab, hairy crab), other aquaculture products (loach, swamp ell, bull frog, clam), dairy products (milk, yogurt) and eggs (chicken egg, duck egg, goose egg, quail egg). A commonly used unit or portion size was specified for each food. Subjects were asked about how often and how many units of food they consumed in a day, a week or a month, and they were also asked about the average volume of water they consumed each day.

### 2.3. Selection and Analysis of Antibiotics in Drinking Water and Animal-Derived Foods

Based on the amount of antibiotics used or detection frequency in foods, drinking water and human urine samples in previous studies, as in Refs. [9,10,12], twenty-one common antibiotics from five categories were selected, including three tetracyclines (tetracycline, oxytetracycline and chlortetracycline), four fluoroquinolones (ciprofloxacin, ofloxacin, norfloxacin and enrofloxacin), five macrolides (azithromycin, roxithromycin, clarithromycin, erythromycin and tilmicosin), six sulfonamides (trimethoprim, sulfadiazine, sulfamethoxazole, sulfamethazine, acetylated sulfamethoxazole and acetylated sulfamethazine) and three phenicols (chloramphenicol, thiamphenicol and florfenicol). Of these, there were four human antibiotics (HAs) exclusively used in humans, four veterinary antibiotics (VAs) exclusively used in animals and thirteen human/veterinary antibiotics (H/VAs) used in both humans and animals [19]. 

Data of antibiotic residues in animal-derived foods were obtained from the notification of unqualified edible agricultural products after special supervision sampling inspection in Deqing County in the last three years (from 2019 to 2021) announced by the Government of Deqing County [20]. The detected items included tetracyclines (tetracycline, oxytetracycline, chlortetracycline, doxycycline), fluoroquinolones (ciprofloxacin, ofloxacin, norfloxacin, enrofloxacin, pefloxacin, sarafloxacin), macrolides (azithromycin, roxithromycin, clarithromycin, erythromycin, tilmicosin, danofloxacin), sulfonamides (sulfadiazine, sulfamethazine, sulphamethazine, sulfamonomethoxine, sulfadimethoxine, sulfamethoxazole, sulfaquinoxaline), phenicols (chloramphenicol, thiamphenicol, florfenicol), metronidazole, and metabolites of furazolidone and furacillin. The selected types of antibiotics analyzed in animal-derived foods were the same as those selected in drinking water.

### 2.4. Collections of Drinking Water Samples and Detections of Antibiotics in Drinking Water

The concentrations of antibiotics in drinking water were mostly around the magnitude of part per billion (ppb), equal to ng/L. Due to the relatively low detection rates of antibiotics in bottled water and barreled water (the corresponding concentrations were less than 0.001 ng/L), as explored in Ref. [9], we only collected tap-water samples in the current study. Based on the geographic distribution, 10 of 194 households were selected to provide a bottle of 2 L tap water as a sample at around 9 a.m. to 10 a.m. for seven consecutive days, respectively (Figure 2). All samples were kept in −20 °C and analyzed within two days. 

A method of ultra-performance liquid chromatography–mass spectrometry (UPLC-MS/MS) was used to measure 21 antibiotics in 70 drinking water samples. Twenty-one antibiotic standard materials and six isotopically labeled internal standards (IS), including tetracycline-d6, ciprofloxacin-d8, azithromycin-d3, sulfamethoxazole-d4, roxithromycin-d7, and chloramphenicol-d5, were purchased from Dr Ehrenstorfer (Augsburg, Germany). After 200 mL of tap water was added by six isotopically labeled antibiotics, and adjusted to pH 6.5–8.5 with formic acid, water samples were passed through 3 mL solid phase extraction (SPE) cartridges packed with 60 mg of hydrophilic-lipophilic balance (HLB) resin at a flow rate of less than 5 mL/min. The HLB cartridges were preconditioned with 2 mL methanol and 2 mL of pure water. After extraction, the cartridge was washed with 2 mL pure water, and then vacuumed for 30 min. The retained antibiotics on the SPE cartridge were eluted with 4 mL of methanol. The eluate was concentrated to dryness at 40 °C with a weak nitrogen flow. The residue was reconstituted in 0.5 mL of a 50% methanol water solution for UPLC–MS/MS analysis. The best separation and organic purge were achieved using 2:3 methanol:acetonitrile (mobile phase A) and 0.2% formic acid water (mobile phase B). The gradient conditions were stated as 90% mobile phase A, held for 1 min, followed by 2% mobile phase B (6 min), 2% mobile phase B (2.5 min), 90% mobile phase B (0.3 min), and finally returned to a 90% mobile phase B (0.5 min). Twenty-four antibiotics were measured in the positive ion mode and three phenicols were measured in the negative ion mode. Parameters were the same for both the positive ion mode and negative ion mode. Nitrogen was used as the nebulizing and desolvation gas, and argon as the collision gas. The method detection limits (MDLs) ranged from 0.004 to 0.296 ng/L. The recovery of 21 antibiotics varied between 70.0 and 120.8%. No interference was found in the field or solvent blanks.

### 2.5. Assessments of Antibiotic Exposures

Animal-derived foods and drinking water are the main dietary routes for human exposure to antibiotics. In order to best present the information on each exposure route, the measure unit was considered as μg/day and ng/day for the animal-derived route and drinking water route, respectively. The overall dietary exposure level to an antibiotic was calculated using Equation (1):(1)$$ADD_{an}=ADD_{an\text{-}food}+ADD_{an\text{-}water}\times 10^{-3}$$
where *ADD_an_*: the average daily dose of overall dietary exposure to an antibiotic via animal-derived foods and drinking water, μg/day; *ADD_an-food_*: the average daily dose of exposure to an antibiotic via animal-derived foods, μg/day; *ADD_an-water_*: the average daily dose of exposure to an antibiotic via drinking water, ng/day.

The exposure level to an antibiotic via animal-derived foods was calculated according to the intake frequency and amount of each food and the detected concentration of this antibiotic in the corresponding food, as Equation (2):(2)$$ADD_{an\text{-}food}=\sum_{1}^{n=i}\left(F_{food\text{-}i}\times M_{food\text{-}i}\times C_{an\text{-}food\text{-}i}\times 10^{-3}\right)$$
where *ADD_an-food_*: the average daily dose of exposure to an antibiotic via animal-derived foods, μg/day; *F_food-i_*: the intake frequency of each food, times/day; *M_food-i_*: the intake amount of each food, g/time; *C_an-food-i_*: the concentration of an antibiotic in the corresponding food, μg/kg.

The exposure level to an antibiotic via drinking water was according the intake volume of drinking water and the detected concentration of this antibiotic in tap water, as Equation (3):(3)$$ADD_{an\text{-}water}=V_{an\text{-}water}\times C_{an\text{-}water}$$
where *ADD_an-water_*: the average daily dose of exposure to an antibiotic via drinking water, ng/day; *V_an-water_*: the volume of drinking water, L/day; *C_an-water_*: the concentration of an antibiotic in tap water, ng/L.

### 2.6. Statistic Analysis

According to the principle of maximum risk control, the maximum detection concentration of each antibiotic was assumed as the contamination level in animal-derived foods and tap water in the prediction of human dietary exposure, so as to avoid possible underestimation [21].

Each category was summed by the contents of antibiotics belonging to this category. Selected percentiles were described for individual antibiotics and antibiotic categories from overall dietary exposure, namely, the animal-derived foods exposure route and drinking water exposure route, respectively. 

The Wilcoxon test and Kruskal–Wallis test were used to examine the differences in the contents of antibiotic exposure among subjects over different sex and age groups. The top ten antibiotics with a greater body burden were identified according to the median and interquartile range (IQR) of the contents. All analyses were performed using R (version 4.0.4, R Foundation for Statistical Computing, Vienna, Austria), and all figures were performed using GraphPad Prism software (version 9, GraphPad Prism, CA, USA). The level of statistical significance was defined as α = 0.05 of a two-side probability.

## 3. Results

### 3.1. Dietary Consumption in This Study Population

The mean age of 523 subjects was 55.17 ± 17.66 years (ranging 18 to 91 years), of whom 269 (51.43%) were female, 465 (88.91%) had less than twelve education years, and 218 (41.68%) were farmers. The median of daily water consumption was 1.25 L (IQR: 0.75–1.75 L) among all subjects. Eggs, livestock meats and freshwater fishes were consumed more frequently, with mean daily consumption probabilities of 56.49%, 46.72% and 37.60%, respectively, while processed meats and animal offal were less often consumed (the daily consumption probabilities were all less than 7%).

### 3.2. Antibiotic Residues in Selected Derived-Foods and Tap Water

Of the twenty-one antibiotics selected in this study, nine antibiotics (tetracycline, oxytetracycline, ofloxacin, norfloxacin, enrofloxacin, trimethoprim, sulfamethazine, chloramphenicol, thiamphenicol, florfenicol) were detected in ten kinds of animal-derived foods, including three livestock meat samples, five poultry samples, one animal offal sample, one processed meat sample, nine shrimp samples, nine freshwater fish samples, three marine fishes, four bull frog samples, one swamp eel sample, and eight egg samples (Appendix A).

Seven antibiotics, including five macrolides (azithromycin, roxithromycin, clarithromycin, erythromycin, tilmicosin) and two sulfonamides (acetylated sulfamethoxazole and acetylated sulfamethazine) were detected in tap water samples, with varying detection frequencies from 2.29% (tilmicosin) to 79.71% (azithromycin). Compared with other antibiotics, tilmicosin (VA) had the highest detected concentration of 6.591 ng/L. Among the three HAs, the maximum detected concentration was 3.993 ng/L for azithromycin, 2.858 ng/L for roxithromycin and 0.849 ng/L for clarithromycin, respectively. Among the three H/VAs, the maximum detected concentration was 1.804 ng/L for erythromycin, 2.992 ng/L for acetylated sulfamethoxazole and 2.307 for acetylated sulfamethazine.

### 3.3. Assessments of Dietary Exposure to Antibiotics

The selected percentiles of the predicted overall dietary exposure level to antibiotics, and those of the predicted exposure level via animal-derived foods or drinking water are shown in Table 1 and Table 2, respectively. Each participant was inevitably exposed to multiple antibiotic residues via two main dietary routes, with an estimated exposure level to a total of sixteen antibiotics ranging from 15.12 to 1128 μg/day (Table 1). The overall dietary exposure level varied greatly in terms of the kinds of antibiotics and sources. Compared with other antibiotics, enrofloxacin contributed to the highest overall dietary exposure level (median: 120.19 μg/day, IQR:8.39–698.78 μg/day), followed by sulfamethazine (median: 32.95 μg/day, IQR: 2.77–162.55 μg/day) and oxytetracycline (median: 28.50 μg/day, IQR: 2.22–146.58 μg/day) (Table 1). All three of these antibiotics were exposed only via the animal-derived foods route, with the maximum exposure level exceeding 100 μg/day (Table 2). In addition, subjects were also simultaneously exposed to five macrolides and two sulfonamides via the drinking water route, but the estimated exposure level to all antibiotics for 50% of subjects was less than 0.03 μg/day (26.74 ng/day) (Table 2).

### 3.4. Overall Dietary Exposure to Antibiotics over Different Sex and Ages

The top ten kinds of antibiotics with the highest median of overall dietary exposure level included enrofloxacin, sulfamethazine, oxytetracycline, tetracycline, trimethoprim, ofloxacin, florfenicol, chloramphenicol, thiamphenicol and tilmicosin, and the majority of these antibiotics showed differences in exposure level by sex and age (Figure 3 and Figure 4). Male participants were more likely to be exposed to antibiotics via dietary routes than female participants, with a median overall dietary exposure level of 290.62 μg/day (IQR: 143.43–470.36 μg/day) and 178.03 μg/day (IQR: 70.87–252.09 μg/day) for males and females, respectively (Figure 3A). The overall dietary exposure levels decreased with age, with a median level of 305.30 μg/day (122.96–360.81 μg/day), 252.10 μg/day (IQR: 117.74–331.40 μg/day) and 182.12 μg/day (IQR: 79.38–289.80 μg/day) for those younger than 35 years, aged 35–60 years and more than 60 years, respectively (Figure 4A). In addition, compared with other kinds of antibiotics, enrofloxacin resulted in a greater body burden of dietary exposure to antibiotics among all subgroups (Figure 3B and Figure 4B).

## 4. Discussion

Generally, humans are exposed to antibiotics via three main sources: medicine use (direct utilization), food consumption and water drinking (contamination by antibiotics). Compared with a short-term exposure to high-level of antibiotics from medicines, humans are more likely to suffer from a long-term but low-level exposure mode in daily diets, which leads to potential health risks [22,23].

In this study, we estimated the daily dietary exposure level to common antibiotics in humans based on a community population in East China. Of the twenty-one antibiotics in the five categories selected, subjects were exposed to a total of sixteen antibiotics, with their amounts ranging from 15.12 to 1128 μg/day via two main dietary routes, including animal-derived foods and drinking water. Consistent with a previous study in Shanghai, presented in Ref. [10], enrofloxacin was listed as the top antibiotic exposed to the study population, of whom 50% were exposed to more than 120 μg/day and a maximum of 698.78 μg/day via daily diets. Enrofloxacin was frequently detected in aquatic products (including freshwater fishes, shrimps, swamp ell, bull frog) in Deqing County with higher detection concentrations, while subjects in this study were more likely to consume freshwater fishes. Recently, enrofloxacin has been widely used in the breeding and production of aquatic products in East China and is considered to an important contaminant to the aquatic environment [24]. Additionally, subjects were exposed to a considerable level of sulfamethazine and oxytetracycline, with 50% of them exposed to more than 32.95 μg/day and 28.50 μg/day via daily diets, respectively.

Given that the median and IQR of the daily exposure level to antibiotics via animal-derived foods was well-matched with those in the overall estimates presented in Table 1, the long-term exposure to low-dose antibiotics in animal-derived foods was inferred to be the primary dietary exposure mode in humans. Consistently, the estimated exposure levels to all antibiotics via drinking water (at the ng/day level, median: 26.74 ng/day, IQR: 16.05–37.44 ng/day) were significantly and substantially lower than those in animal-derived foods (at the μg/day level, median: 216.38 μg/day, IQR: 87.52–323.00 μg/day). Such findings indicated that drinking water was an inevitable source of some antibiotics, but played a limited role in the total dietary exposure to antibiotics [6]. Moreover, the overall dietary exposure levels of antibiotics showed differences by sex and age, which might be related to dietary habits and the levels of antibiotic residues in animal-derived foods and drinking water. Younger people were more likely to be exposed to antibiotics via the dietary route than other age subgroups, mostly due to their higher level of food consumption [25].

Relatively few studies have assessed human exposure to antibiotics in daily diets. Two studies in Shanghai estimated the daily exposure of adults to antibiotic residues in animal-derived foods, and the daily exposure of children to antibiotic residues in tap water, respectively, using Monte Carlo Simulation to estimate the consumption of foods and water due to a lack of accurate consumption data [9,10] One of these studies reported that the subjects were exposed to 15 of 20 selected antibiotics, and the estimated exposure level via meats and aquatic products ranged from 0 to 0.111 μg/kg/day for 95% male and ranged from 0 to 0.125 μg/kg/day for 95% female, respectively, according to Ref. [10]. These exposure levels were far below the results in our study (a range of 0.40 to 14.78 μg/kg/day for 95% male and 0.27 to 6.71 μg/kg/day for 95% female, respectively), if based on the similar distribution of body weight, for which we had a mean of 63.8 kg (SD 9.9 kg) for male and 56.5 kg (SD 9.1 kg) for females according to the data provided by Group of China Obesity Task Force [26]. By contrast, another study reported that the subjects were exposed to only 2 of 21 selected antibiotics, and the estimated exposure level via tap water was less than 1 ng/kg/day in 95% children, which can be found in Ref. [9]. These exposure levels were comparable to the results in our study (a range of 0.1 to 1.5 ng/kg/day for 95% male and 0.09 to 0.9 ng/kg/day for 95% female, respectively).

Antibiotic residues in animal-derived foods, such as meats, aquatic products and dairy products, result from their applications as veterinary drugs and feed additives in animal husbandry and the aquaculture industry, especially in the intensive cultivation circumstances [27]. A substantial level of antibiotic residues were detected in animal-derived foods in Deqing County, as well as some human antibiotics or veterinary drugs banned according to relevant regulations [28]. There is no doubt that an inappropriate use of antibiotics in animal husbandry and the aquaculture industry is common. Humans are therefore likely to consume these animal-derived foods that contain substantial residues of these antibiotics.

The primary source of antibiotics in aquatic environments is the excretion of antibiotics from humans and animals, via untreated sewage [29], medical waste [30] and aquaculture discharge [31]. Fluoroquinolones, phenicols, sulfonamides and macrolides have been frequently detected in the surface water of Yangtze River and Taihu Lake [32,33]. In our study, we found three macrolides (azithromycin, roxithromycin and erythromycin) and two sulfonamides (acetylated sulfamethoxazole and acetylated sulfamethazine) in tap water samples, which is somewhat consistent with a few studies [34]. Two phenicols (florfenicol and thiamphenicol) have been previously found in terminal tap water in Shanghai, which also relies on the Yangtze River as a source of drinking water [9]. Such evidence suggested that water-treatment processing in East China should be improved to remove some antibiotic residues to reduce potential health risks, although all detected antibiotics were under the standard limits for tap water in China.

To the best of our knowledge, this study was the first to comprehensively assess daily dietary exposure to antibiotics in humans in East China. Human dietary exposure to antibiotics in this study was more accurately estimated by combining the data of antibiotic contaminations in foods and drinking water, and the information of daily dietary consumption, compared to estimates in others studies, which studied either a single source (from either foods or drinking water) or used an estimate of dietary consumption. However, some limitations should be highlighted. First, we assumed the same contamination level in tap water as that in drinking water, and failed to consider the degradation of antibiotics due to some cooking processes, all of which would mean we overestimated the true exposure contents. Second, the data of antibiotic residues in animal-derived foods were obtained from the official website of the People’s Government of Deqing County, instead of via our own testing, therefore, the contamination level of antibiotics in the foods reported may be inconsistent with those consumed by our subjects, so there might be some systematic errors. Third, dietary consumption was self-reported by subjects, which means the study may suffer from report bias.

In conclusion, considering antibiotics residues in foods and drinking water may lead to an important and inadvertent dietary exposure, our study suggests that it is essential to implement a rigorous policy to control the use of antibiotics, or explore some alternatives for antibiotics, in order to reduce the residual levels of antibiotics in foods and drinking water. Our study also provides a scientific basis for the market monitoring and risk management of antibiotics.

## 5. Conclusions

The community population investigated in East China was extensively exposed to multiple antibiotics via dietary routes. The long-term exposure to low-dose antibiotics in animal-derived foods was the primary dietary exposure mode, compared with drinking water. Enrofloxacin contributed to the major body burden of dietary exposure, based on the combination of the consumption of aquatic products and considerable enrofloxacin residues in them. Although the human dietary exposure level to antibiotics via drinking water and animal-derived foods ranged from ng/day to μg/L, their chronic toxicity and the accumulation and spread of ABR may be potential hazards to humans. Therefore, long-term monitoring of antibiotic contaminations in foods and drinking water, and human dietary antibiotic exposure is warranted.

## Figures and Tables

**Figure 1 antibiotics-11-00407-f001:**
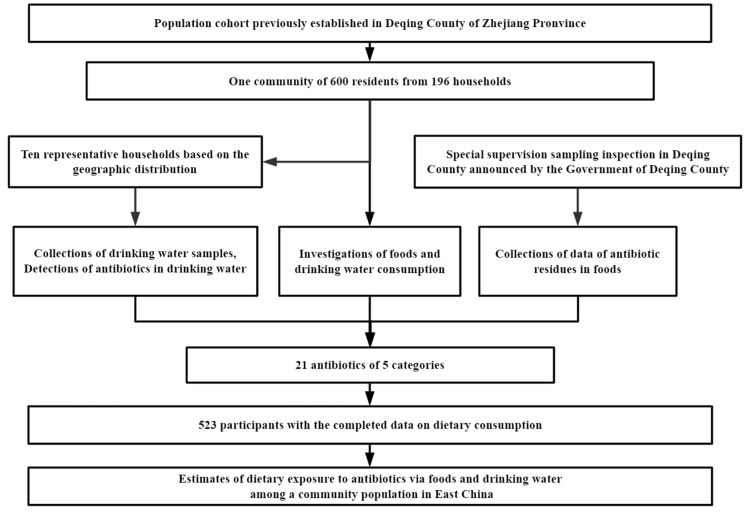
Flow chart for the study.

**Figure 2 antibiotics-11-00407-f002:**
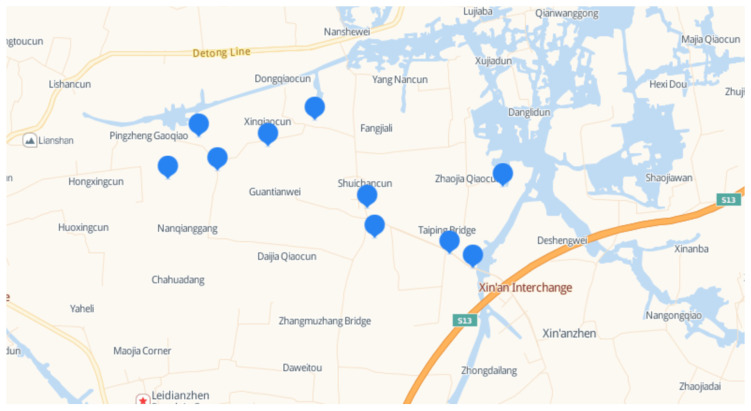
Collection sites of drinking water.

**Figure 3 antibiotics-11-00407-f003:**
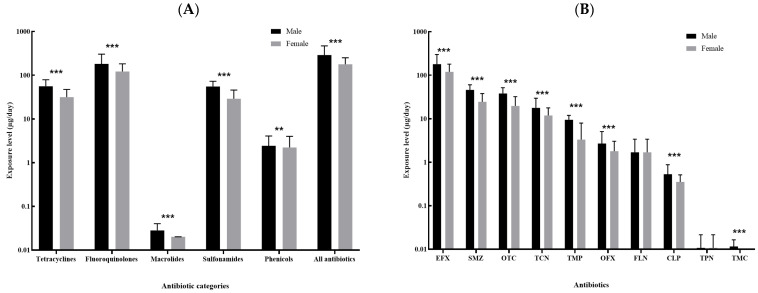
Comparison of the dietary exposure contents to antibiotics between male and female: (**A**). the five categories of antibiotics and the sum of all antibiotics; (**B**). the top ten kinds of antibiotics with the highest median contents (EFX: Enrofloxacin; SMZ, Sulfamethazine; OTC: Oxytetracycline; TCN, Tetracycline; TMP: Trimethoprim; OFX: Ofloxacin; FLN: Florfenicol; CLP: Chloramphenicol; TPN, Thiamphenicol; TMC, Tilmicosin.). ** 0.05 < *p* < 0.01; *** *p* < 0.001.

**Figure 4 antibiotics-11-00407-f004:**
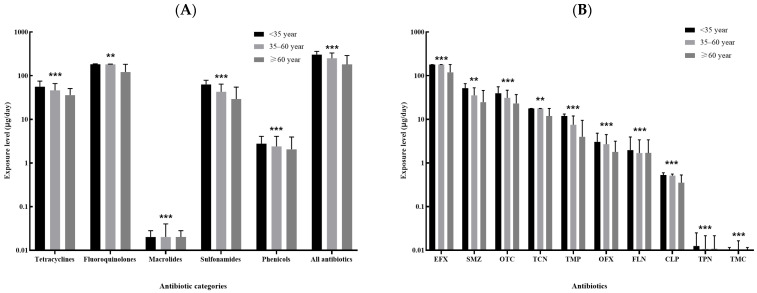
Comparison of the dietary exposure contents to antibiotics between subjects less than 35 years, 35–60 years and more than 60 years: (**A**). the five categories of antibiotics and the sum of all antibiotics; (**B**). the top ten kinds of antibiotics with the highest median of contents (EFX: Enrofloxacin; SMZ, Sulfamethazine; OTC: Oxytetracycline; TCN, Tetracycline; TMP: Trimethoprim; OFX: Ofloxacin; FLN: Florfenicol; CLP: Chloramphenicol; TPN, Thiamphenicol; TMC, Tilmicosin.). ** 0.05 < *p* < 0.01; *** *p* < 0.001.

**Table 1 antibiotics-11-00407-t001:** Overall dietary exposure level to antibiotics among subjects.

Antibiotics	Usage ^a^	Overall Dietary Exposure Level (μg/day)								
		Minimum	Percentiles							Maximum
			2.5th	5th	25th	50th	75th	95th	97.5th	
Tetracyclines ^b^		3.05	4.57	6.84	24.05	40.77	62.33	104.23	128.18	215.70
Tetracycline	H/VA	0.83	0.83	1.24	4.15	11.89	17.83	35.67	58.55	69.13
Oxytetracycline	H/VA	2.22	3.34	5.01	17.33	28.50	44.78	70.71	82.85	146.58
Chlortetracycline	VA	-	-	-	-	-	-	-	-	-
Fluoroquinolones ^b^		8.61	8.61	12.91	42.67	123.35	185.02	369.83	600.65	717.15
Ciprofloxacin	H/VA	-	-	-	-	-	-	-	-	-
Ofloxacin	H/VA	0.22	0.22	0.33	0.74	2.69	3.72	7.28	9.16	18.38
Norfloxacin	H/VA	-	-	-	-	-	-	-	-	-
Enrofloxacin	VA	8.39	8.39	12.58	41.93	120.19	180.28	360.57	591.86	698.78
Macrolides ^b^		<0.01	<0.01	<0.01	0.01	0.02	0.03	0.06	0.06	0.06
Azithromycin	HA	<0.01	<0.01	<0.01	<0.01	<0.01	0.01	0.02	0.02	0.02
Roxithromycin	HA	<0.01	<0.01	<0.01	<0.01	<0.01	0.01	0.01	0.01	0.01
Clarithromycin	HA	<0.01	<0.01	<0.01	<0.01	<0.01	<0.01	<0.01	<0.01	<0.01
Erythromycin	H/VA	<0.01	<0.01	<0.01	<0.01	<0.01	<0.01	0.01	0.01	0.01
Tilmicosin	VA	<0.01	<0.01	<0.01	<0.01	0.01	0.01	0.03	0.03	0.03
Sulfonamides ^b^		3.33	5.89	8.01	20.08	39.79	63.12	100.98	123.31	200.25
Trimethoprim	H/VA	0.56	0.83	0.83	2.78	6.61	11.96	19.20	22.72	37.69
Sulfadiazine	H/VA	-	-	-	-	-	-	-	-	-
Sulfamethoxazole	H/VA	-	-	-	-	-	-	-	-	-
Sulfamethazine	H/VA	2.77	5.05	6.78	17.77	32.95	51.47	84.05	101.2	162.55
Acetylated sulfamethoxazole	H/VA	<0.01	<0.01	<0.01	<0.01	<0.01	0.01	0.01	0.01	0.01
Acetylated sulfamethazine	H/VA	<0.01	<0.01	<0.01	<0.01	<0.01	<0.01	0.01	0.01	0.01
Phenicols ^b^		0.15	0.25	0.47	1.83	2.26	4.03	4.66	5.01	6.04
Chloramphenicol	HA	0.03	0.03	0.04	0.14	0.37	0.55	1.08	1.70	2.07
Thiamphenicol	H/VA	<0.01	<0.01	<0.01	0.01	0.01	0.02	0.03	0.03	0.03
Florfenicol	VA	0.12	0.12	0.12	1.70	1.70	3.39	3.95	3.95	3.95
All antibiotics ^c^		15.12	23.46	31.31	87.52	216.38	323.00	572.68	816.08	1128.00

-: No value or below limit of detection; ^a^ Usage: VA, veterinary antibiotic exclusively used in animal; HA, human antibiotic exclusively used in humans; H/VA, human/veterinary antibiotic used in both animals and humans. ^b^ Sum of contents of all antibiotics in the corresponding category; ^c^ Sum of contents of all antibiotics selected.

**Table 2 antibiotics-11-00407-t002:** Dietary exposure level to antibiotics via animal-derived foods and drinking water among subjects.

Antibiotics	Usage ^a^	Derived-Foods									Drinking Water								
		Exposure Level (μg/day)									Exposure Level (ng/day)								
		Minimum	Percentiles							Maximum	Minimum	Percentiles							Maximum
			2.5th	5th	25th	50th	75th	95th	97.5th			2.5th	5th	25th	50th	75th	95th	97.5th	
Tetracyclines ^b^		3.05	4.57	6.84	24.05	40.77	62.33	104.23	128.18	215.70	-	-	-	-	-	-	-	-	-
Tetracycline	H/VA	0.83	0.83	1.24	4.15	11.89	17.83	35.67	58.55	69.13	-	-	-	-	-	-	-	-	-
Oxytetracycline	H/VA	2.22	3.34	5.01	17.33	28.50	44.78	70.71	82.85	146.58	-	-	-	-	-	-	-	-	-
Chlortetracycline	VA	-	-	-	-	-	-	-	-	-	-	-	-	-	-	-	-	-	-
Fluoroquinolones ^b^		8.61	8.61	12.91	42.67	123.35	185.02	369.83	600.65	717.15	-	-	-	-	-	-	-	-	-
Ciprofloxacin	H/VA	-	-	-	-	-	-	-	-	-	-	-	-	-	-	-	-	-	-
Ofloxacin	H/VA	0.22	0.22	0.33	0.74	2.69	3.72	7.28	9.16	18.38	-	-	-	-	-	-	-	-	-
Norfloxacin	H/VA	-	-	-	-	-	-	-	-	-	-	-	-	-	-	-	-	-	-
Enrofloxacin	VA	8.39	8.39	12.58	41.93	120.19	180.28	360.57	591.86	698.78	-	-	-	-	-	-	-	-	-
Macrolides ^b^		-	-	-	-	-	-	-	-	-	4.02	4.02	4.02	12.07	20.12	28.17	61.97	64.38	64.38
Azithromycin	HA	-	-	-	-	-	-	-	-	-	1.00	1.00	1.00	2.99	4.99	6.99	15.37	15.97	15.97
Roxithromycin	HA	-	-	-	-	-	-	-	-	-	0.71	0.71	0.71	2.14	3.57	5.00	11.00	11.43	11.43
Clarithromycin	HA	-	-	-	-	-	-	-	-	-	0.21	0.21	0.21	0.64	1.06	1.49	3.27	3.40	3.40
Erythromycin	H/VA	-	-	-	-	-	-	-	-	-	0.45	0.45	0.45	1.35	2.26	3.16	6.95	7.22	7.22
Tilmicosin	VA	-	-	-	-	-	-	-	-	-	1.65	1.65	1.65	4.94	8.24	11.53	25.38	26.36	26.36
Sulfonamides ^b^		3.32	5.88	8.01	20.07	39.79	63.11	100.97	123.30	200.24	1.32	1.32	1.32	3.97	6.62	9.27	20.40	21.20	21.20
Trimethoprim	H/VA	0.56	0.83	0.83	2.78	6.61	11.96	19.2	22.72	37.69	-	-	-	-	-	-	-	-	-
Sulfadiazine	H/VA	-	-	-	-	-	-	-	-	-	-	-	-	-	-	-	-	-	-
Sulfamethoxazole	H/VA	-	-	-	-	-	-	-	-	-	-	-	-	-	-	-	-	-	-
Sulfamethazine	H/VA	2.77	5.05	6.78	17.77	32.95	51.47	84.05	101.2	162.55	-	-	-	-	-	-	-	-	-
Acetylated sulfamethoxazole	H/VA	-	-	-	-	-	-	-	-	-	0.75	0.75	0.75	2.24	3.74	5.24	11.52	11.97	11.97
Acetylated sulfamethazine	H/VA	-	-	-	-	-	-	-	-	-	0.58	0.58	0.58	1.73	2.88	4.04	8.88	9.23	9.23
Phenicols ^b^		0.15	0.25	0.47	1.83	2.26	4.03	4.66	5.01	6.04	-	-	-	-	-	-	-	-	-
Chloramphenicol	HA	0.03	0.03	0.04	0.14	0.37	0.55	1.08	1.70	2.07	-	-	-	-	-	-	-	-	-
Thiamphenicol	H/VA	<0.01	<0.01	<0.01	0.01	0.01	0.02	0.03	0.03	0.03	-	-	-	-	-	-	-	-	-
Florfenicol	VA	0.12	0.12	0.12	1.70	1.70	3.39	3.95	3.95	3.95	-	-	-	-	-	-	-	-	-
All antibiotics ^c^		15.12	23.46	31.31	87.52	216.38	323.00	572.68	816.08	1128.00	5.35	5.35	5.35	16.05	26.74	37.44	82.37	85.58	85.58

-: No value or below limit of detection; ^a^ Usage: VA, veterinary antibiotic exclusively used in animal; HA, human antibiotic exclusively used in humans; H/VA, human/veterinary antibiotic used in both animals and humans. ^b^ Sum of contents of all antibiotics in the corresponding category; ^c^ Sum of contents of all antibiotics selected.

## Data Availability

The datasets for this manuscript will be made available upon request pending, further inquiries can be directed to the corresponding author Na Wang, na.wang@fudan.edu.cn or Chaowei Fu, fcw@fudan.edu.cn.

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
