# Peer review of "Estimates of Dietary Exposure to Antibiotics among a Community Population in East China"

_antibiotics, 2022, doi:10.3390/antibiotics11030407_

Round 1

Reviewer 1 Report

The manuscript is scientifically more or less sound. The goal of the paper, to look into the mostly neglected exposure of consumers to antibiotics through consuming food and water that are contaminated with antibiotic residues, is well described. Material and Methods are comprehensible as well as the Results. The conclusion in the end of the text is the same two short sentences as in the abstract, and it should go into more detail: there is more to conclude - e.g. a recommendation to enforce the testing of slaughter carcasses, milk and aquacultural food for antibiotic residues before consumption. In the European Union, there has been for decades a very strict and strongly supervised European control plan for residue testing.

With some reference to international experiences and data, the discussion and the clonclusion would be remarkably improved.

As for the langauge: There are several sentences that have mostly grammar deficiencies:

Line 35: "Community population were..." - Correct is: The community population that was investigated was...

Line 51: "In 2018, the usage rate of ... was high as 40.4%" - Percent of what? And should it not be "was

Line 54: "..a high detection rate of ABR and become..." -  Correct is: has become...

Lines 66/67: "....in human"  - Correct is: in humansas high as 40.4%?

The same mistake is in Lines 68 and 69.

In Figure 1 in the first box: "prebiously" must be "previously"

Line 77: "Those were excluded with serious...."   should be: Those with serious... were excluded

These are only examples of incorrect language passages. Please look for somebody to proofread the English langauge throughout the entire manuscript

Author Response

Dear reviewer,

Thank you for taking the time and effort to review our manuscript (ID: antibiotics-1620202) entitled “Estimates of dietary exposure to antibiotics among a community population in East China ”. We appreciate your constructive comments and have revised the manuscript carefully in order to meet your requirements. We have uploaded the revised marked manuscript with all the changes marked up using the “Track Changes” function. Appended to this letter is our point-by-point response to the comments raised by the editors and reviewers. The comments are listed and our responses are given directly afterward.

The manuscript is scientifically more or less sound. The goal of the paper, to look into the mostly neglected exposure of consumers to antibiotics through consuming food and water that are contaminated with antibiotic residues, is well described. Material and Methods are comprehensible as well as the Results.

  1. The conclusion in the end of the text is the same two short sentences as in the abstract, and it should go into more detail: there is more to conclude - e.g. a recommendation to enforce the testing of slaughter carcasses, milk and aqua-cultural food for antibiotic residues before consumption. In the European Union, there has been for decades a very strict and strongly supervised European control plan for residue testing.With some reference to international experiences and data, the discussion and the conclusion would be remarkably improved.

Response: Thank you for your suggestions. We have expanded the introduction, discussions, conclusions and corresponding abstract. Detailed contents are presented as follows:

  • Introduction: in Line 60-69 and 76-88, Page 2.
  • Discussions: in Line 150-155, Page 14.
  • Conclusions: in Line 159-168, Page 14.

  1. As for the language: There are several sentences that have mostly grammar deficiencies:

Line 35: "Community population were..." - Correct is: The community population that was investigated was...

Response: “Community population were...” has been changed to “The community population investigated in East China were extensively exposed to...” in Line 36-37, Page 1.

Line 51: "In 2018, the usage rate of ... was high as 40.4%" - Percent of what? And should it not be "was as high as 40.4%?"

Response: The usage percentage of antibiotics in inpatients is presented as the number of inpatients using antibiotics divided by the total number of hospitalized patients during the same period. “the usage rate of ” has been changed to “the usage percentage of ” in Line 59, Page 2.

Line 54: "..a high detection rate of ABR and become..." -  Correct is: has become...

Response: This sentence has been changed to “East China has become an important region affected by the significantly increasing consumption of antibiotics, and become the hot spot of controlling antibiotic use and ABR” in Line 67-69, Page 2.

Lines 66/67: "....in human"  - Correct is: in humans 

The same mistake is in Lines 68 and 69.

Response: Similar mistakes have been corrected in Line 14, 16, Page1 and Line 59, 79, 93-95, Page 2.

In Figure 1 in the first box: "prebiously" must be "previously"

Response: This word has been corrected in Figure 1.

Line 77: "Those were excluded with serious...."   should be: Those with serious... were excluded

Response: This sentence has been changed to “Those with serious body diseases such as cardiovascular and cerebrovascular diseases, liver and kidney diseases were excluded.” in Line 104-105, Page 3.

These are only examples of incorrect language passages. Please look for somebody to proofread the English language throughout the entire manuscript.

Response: The whole manuscript has been proofread and revised, and some mistakes have been corrected.

Reviewer 2 Report

The study proposed by Authors aims to evaluate the total intake level of antibiotics in human from foods and drinking water based on a community population in East China. The Authors have interviewed  600 local residents from 194 households in Deqing County of Zhejiang Province. The dietary exposure to antibiotics were estimated by combining the data of antibiotic contamination in foods and drinking water, and the information of dietary consumption and proving that community population were very exposed to multiple antibiotics.

The study is interesting but the introduction should be expanded.

Line 24 delete "in human"

Author Response

Dear reviewer,

Thank you for taking the time and effort to review our manuscript (ID: antibiotics-1620202) entitled “Estimates of dietary exposure to antibiotics among a community population in East China ”. We appreciate your constructive comments and have revised the manuscript carefully in order to meet your requirements. We have uploaded the revised marked manuscript with all the changes marked up using the “Track Changes” function. Appended to this letter is our point-by-point response to the comments raised by the editors and reviewers. The comments are listed and our responses are given directly afterward.

Author response:

The study proposed by Authors aims to evaluate the total intake level of antibiotics in human from foods and drinking water based on a community population in East China. The Authors have interviewed 600 local residents from 194 households in Deqing County of Zhejiang Province. The dietary exposure to antibiotics were estimated by combining the data of antibiotic contamination in foods and drinking water, and the information of dietary consumption and proving that community population were very exposed to multiple antibiotics.

  1. The study is interesting but the introduction should be expanded.

Response: Thank you for your suggestions. We have expanded the introduction, detailed contents are presented in Line 60-69 and 76-88, Page 2.

  1. Line 24 delete "in human".

Response: These words have been deleted.

Reviewer 3 Report

The manuscript presents an estimation of the amount of antibiotics people, from a small community in East China, are exposed to trough consummation of food and drinking water. Extensive use of antibiotics in agriculture and aquaculture as well as in medicine in general has resulted in an overexposure to different kinds of antibiotics causing many risks to human health. The topic is therefore an interesting one but there are some lacks in the methodology of the manuscript. Authors use specific calculations to asses the exposure to specific antibiotics by consuming specific types of food in amounts provided by investigated subjects but there is no clear evidence of the content of antibiotics in the food those subjects really consume. Table S1 provides only partial information on some mentioned antibiotics in general food types. The main idea is an “estimation” but even an estimation should be based on some concrete evidence/information.

The methodology is not clearly presented. The sections in Material and methods should be better organized e.g. 2.1. should have information about collection sites of drinking water which are mentioned in the flow chart in this chapter. Sections 2.3. and 2.5 could be merged together. In 2.4. detailed information on how was the UPLC-Q/TOF performed should be provided.

There are some difficulties in following the text due to the problems in English language. It is often not clear what is exposed to what… e.g. lines 211-212 (Page 6), lines 34-36 and 41-43 (Page 10) etc..

Figure 3. is the same as figure 2. In figure description the chosen defined statistical signivicance value should be stated.

References in the text should be divided by comma.

Author Response

Dear reviewer,

Thank you for taking the time and effort to review our manuscript (ID: antibiotics-1620202) entitled “Estimates of dietary exposure to antibiotics among a community population in East China ”. We appreciate your constructive comments and have revised the manuscript carefully in order to meet your requirements. We have uploaded the revised marked manuscript with all the changes marked up using the “Track Changes” function. Appended to this letter is our point-by-point response to the comments raised by the editors and reviewers. The comments are listed and our responses are given directly afterward.

Author response:

The manuscript presents an estimation of the amount of antibiotics people, from a small community in East China, are exposed to trough consummation of food and drinking water. Extensive use of antibiotics in agriculture and aquaculture as well as in medicine in general has resulted in an overexposure to different kinds of antibiotics causing many risks to human health. The topic is therefore an interesting one but there are some lacks in the methodology of the manuscript.

  1. Authors use specific calculations to assessthe exposure to specific antibiotics by consuming specific types of food in amounts provided by investigated subjects but there is no clear evidence of the content of antibiotics in the food those subjects really consume. Table S1 provides only partial information on some mentioned antibiotics in general food types. The main idea is an “estimation” but even an estimation should be based on some concrete evidence/information.

Response:

Similar estimation has been made in previous studies, which using the data of Cadmium (Cd) contamination from literature, to systematically analyze the dietary Cd exposure of Shanghai residents from 1988 to 2018[1]. Reference:

[1] Qing Y, Yang J, Zhu Y, et al. Cancer risk and disease burden of dietary cadmium exposure changes in Shanghai residents from 1988 to 2018. Sci Total Environ. 2020;734:139411.

    In our study, the data of antibiotic residues in animal-derived foods were obtained from the official website of the People’s Government of Deqing County, instead of via our own testing, the contamination level of antibiotics in foods reported may be inconsistent with those consumed by our subjects, so there might be some systematic errors. Above sentences have been added into the Limitation in Line 144-148, Page 13-14.

Although Table S1 just provides only partial information on selected antibiotics and contaminated foods, specific antibiotic residues have been frequently detected in these types of foods, and their residue levels were closely related with their usage amount in livestock industry. In addition, these foods were collected from several larger fresh markets and supermarkets in Deqing, we assumed residents in Deqing County (including our subjects) were more likely to consume these foods.

   In general, the results in our study may indicate that antibiotics residues in foods and drinking water may lead to an important and inadvertent dietary exposure, and provide a scientific basis for the market monitoring and risk management on antibiotics.

  1. The methodology is not clearly presented. The sections in Material and methods should be better organized e.g. 2.1. should have information about collection sites of drinking water which are mentioned in the flow chart in this chapter. Sections 2.3. and 2.5 could be merged together. In 2.4. detailed information on how was the UPLC-Q/TOF performed should be provided.

Response: Thank you for your suggestions.

(1)The map of sampling sites has been shown in Figure 2.

(2)Section 2.3 and Section 2.5 have been merged into the current Section 2.3.

(3)Detection information of selected antibiotics in drinking water has been added in Section 2.4 in Line 166-187, Page 4-5.

  1. There are some difficulties in following the text due to the problems in English language. It is often not clear what is exposed to what… e.g. lines 211-212 (Page 6), lines 34-36 and 41-43 (Page 10) etc..

Response:

(1)These mentioned sentences have been proofread and revised in Line 286-289 (Page 7), Line 70-73, 73-75 and 80-82 (Page 12).

(2)Definitions of dietary exposure in our study included three parts:

①Overall dietary exposure, namely exposure via daily diets, including the combination of animal-derived foods and drinking water ;

②Dietary exposure via only animal-derived foods ;

③Dietary exposure via only drinking water.

  1. Figure 3. is the same as figure 2. In figure description the chosen defined statistical significance value should be stated.

Response: Figure 2 has been corrected, and the description of statistical significance value has been added in the figure title.

  1. References in the text should be divided by comma.

Response: Citation formats have been corrected.

Reviewer 4 Report

The submitted manuscript is prepared well and a valuable contribution. The introduction sufficiently lead to the topic, however it could be extended with more information regarding lines 53-54. The experiment design is clear and accurately described. Nevertheless, more information regarding determination of antibiotic in water should be provided, including source and purity of materials. Results are clearly presented, however in some places including abstract it should be more clearly stated that numbers relate to daily exposure. Discussion is proper and relevant. Citations in the text should be adopted to journal requirements.

Author Response

Dear reviewer,

Thank you for taking the time and effort to review our manuscript (ID: antibiotics-1620202) entitled “Estimates of dietary exposure to antibiotics among a community population in East China ”. We appreciate your constructive comments and have revised the manuscript carefully in order to meet your requirements. We have uploaded the revised marked manuscript with all the changes marked up using the “Track Changes” function. Appended to this letter is our point-by-point response to the comments raised by the editors and reviewers. The comments are listed and our responses are given directly afterward.

Author response:

The submitted manuscript is prepared well and a valuable contribution.

  1. The introduction sufficiently lead to the topic, however it could be extended with more information regarding lines 53-54.

Response: Thank you for your suggestions.We have expanded the introduction, detailed contents are presented in Line 60-69 and 76-88, Page 2.

  1. The experiment design is clear and accurately described. Nevertheless, more information regarding determination of antibiotic in water should be provided, including source and purity of materials.

Response: We have improved the Section Method in the following parts:

(1)The map of sampling sites has been shown in Figure 2.

(2)Section 2.3 and Section 2.5 have been merged into the current Section 2.3.

(3)Detection information of antibiotics in drinking water has been added in Section 2.4,

Line 166-187, Page 4-5.

  1. Results are clearly presented, however in some places including abstract it should be more clearly stated that numbers relate to daily exposure.

Response:

(1)The unit for exposure level has been changed to μg/day or ng/day in order to well present the daily exposure level. And formula for calculating dietary exposure have been provided more clearly in Section 2.5, Line 211-238, Page 5-6.

(2)Corresponding descriptions have been improved in Abstract (Line 25-36, Page 1) and Results (Line 283-297, Page 7 and Line 1-14, Page 11).

  1. Discussion is proper and relevant. Citations in the text should be adopted to journal requirements.

Response: Citation formats have been corrected.

Round 2

Reviewer 1 Report

The mendments according to the reviewer's recommenadtions are well taken.

In Figure 1, on line 123 there is still "prviously" written as "prebiously". Please correct this.

A final proof-reading of the new text passages is recommended.

Reviewer 3 Report

The authors have well addresed all the comments from the previous revision. Aditionall proofreading of the English language is recommended.